# Structures, Mechanisms, and Physiological Functions of Zinc Transporters in Different Biological Kingdoms

**DOI:** 10.3390/ijms25053045

**Published:** 2024-03-06

**Authors:** Han Ba Bui, Kenji Inaba

**Affiliations:** 1Institute of Multidisciplinary Research for Advanced Materials, Tohoku University, Sendai 980-8577, Japan; bui.ba.han.e2@tohoku.ac.jp; 2Department of Molecular and Chemical Life Sciences, Graduate School of Life Sciences, Tohoku University, Sendai 980-8577, Japan; 3Medical Institute of Bioregulation, Kyushu University, Fukuoka 812-8582, Japan; 4Department of Chemistry, Graduate School of Science, Tohoku University, Sendai 980-8578, Japan; 5Core Research for Evolutional Science and Technology (CREST), Japan Agency for Medical Research and Development (AMED), Chiyoda-ku, Tokyo 100-0004, Japan

**Keywords:** zinc transporter, ZnT, YiiP, histidine-rich loop, cryo-EM

## Abstract

Zinc transporters take up/release zinc ions (Zn^2+^) across biological membranes and maintain intracellular and intra-organellar Zn^2+^ homeostasis. Since this process requires a series of conformational changes in the transporters, detailed information about the structures of different reaction intermediates is required for a comprehensive understanding of their Zn^2+^ transport mechanisms. Recently, various Zn^2+^ transport systems have been identified in bacteria, yeasts, plants, and humans. Based on structural analyses of human ZnT7, human ZnT8, and bacterial YiiP, we propose updated models explaining their mechanisms of action to ensure efficient Zn^2+^ transport. We place particular focus on the mechanistic roles of the histidine-rich loop shared by several zinc transporters, which facilitates Zn^2+^ recruitment to the transmembrane Zn^2+^-binding site. This review provides an extensive overview of the structures, mechanisms, and physiological functions of zinc transporters in different biological kingdoms.

## 1. Introduction

Zinc ions (Zn^2+^), an essential trace element in bacteria, fungi, plants, and animals, including humans [1], serve as a key component in many signal transduction processes and act as an essential cofactor for many proteins and enzymes [2,3]. Zinc deficiency causes several human diseases [4,5,6,7,8,9,10,11,12,13]; indeed, zinc supplements have beneficial effects on human health [8,14,15,16,17,18,19]. However, excessive adsorption of Zn^2+^ leads to disruption of the gastrointestinal flora balance, deficiency of other essential heavy metals, including iron, copper, and manganese, and reduction in immune function [20,21,22,23]. Zn^2+^ also plays an important role in the physiology of organisms such as plants and bacteria [24,25]. In plants, zinc deficiency is linked to growth defects and inhibition of flowering [26,27]. Additionally, Zn^2+^ is responsible for the virulence of some bacteria [28]. Since Zn^2+^ is involved in numerous biological events, humans, plants, yeasts, and bacteria have evolved elaborate Zn^2+^ transport systems that respond to Zn^2+^ perturbation.

Failure of the Zn^2+^ transport systems plays a role in diseases such as cancer [29,30], Alzheimer’s [31,32], and Parkinson’s [33,34], as well as temporary zinc deficiency in newborns [35], perinatal fatal cardiomyopathy [36], risk of febrile seizures [37], Lowe’s syndrome [38], disorders of muscle tone with polycythemia [39,40], and chronic liver disease [40]. Therefore, human zinc transporters (ZnTs) are potential targets of drugs and preclinical diagnostic tests. Owing to the important physiological roles, and pharmacological and preclinical diagnostic significance of Zn^2+^ transport systems, a variety of biochemical, structural, physiological, and genetic experiments have been carried out over the past several decades to better understand their functions and mechanisms. The most comprehensively studied bacterial zinc transporter is YiiP, which works in *Escherichia coli* and *Shewanella oneidensis* (EcYiiP and SoYiiP, respectively) [41,42,43,44,45,46,47,48,49]. These transporters are a convenient model to study the general mechanisms underlying Zn^2+^ transport. The most intensively studied mammalian ZnTs are SLC30A7/ZnT7 [50] and SLC30A8/ZnT8 [51,52]. Our interests in ZnT family members stem mainly from their roles in maintaining Zn^2+^ homeostasis in cellular organelles throughout the body and the fact that their dysfunction causes serious diseases.

As is the case for other membrane transporters, ZnTs undergo conformational conversion to transport Zn^2+^ across biological membranes. To fully understand the mechanism underlying Zn^2+^ transport, high-resolution structures of the transporters have been captured in different states. The first X-ray crystal structure of a zinc transporter (Table 1) was reported for EcYiiP [41,42], followed by the EM structure of SoYiiP [43,44,45,46]. More recently, cryo-EM structures of vertebrate ZnTs have been reported (Table 1); these include *Homo sapiens* ZnT7 (HsZnT7) [50], *Homo sapiens* ZnT8 (HsZnT8) [51], and *Xenopus tropicalis* ZnT8 (XtZnT8) [52]. These structures allow us to propose an updated model of ZnTs-mediated Zn^2+^ transport. Of note, our recent structural and biochemical studies on HsZnT7 revealed the role of its cytosolic histidine-rich loop (His-loop) in efficient Zn^2+^ uptake [50]. Thus, we have built on the structural and mechanistic foundations of ZnTs in the biological kingdom, while making significant progress regarding research into other members with Zn^2+^ transport functions.

## 2. Zn^2+^ Transport Systems in Prokaryotes and Eukaryotes

Prokaryotes and eukaryotes have developed a variety of Zn^2+^ transport systems to promote the uptake or efflux of Zn^2+^ across biological membranes. ZnTs can be divided into three major groups depending on the mode of transport: Uniporters that transport Zn^2+^ alone; symporters that transport Zn^2+^ in the same direction as other ions, such as protons; and antiporters that transport Zn^2+^ and another ion in opposite directions, such that the binding of one is concomitant with the release of the other. In general, uniporters require no external energy input and transport specific molecules along their concentration gradients; they are therefore passive transporters. However, it can also act as an active transporter if the transport process is against the concentration gradient. By contrast, symporters and antiporters use the energy stored in the concentration gradient of another ion, in many cases, a proton, to transport specific molecules against their concentration gradients. In this regard, symporters and antiporters can be regarded as active transporters. In addition, some P-ATPases and ABC transporters transport Zn^2+^ using ATP as an external energy source to overcome the Zn^2+^ concentration gradient.

Zinc transporters (ZnTs) and ZRT- and IRT-related proteins (ZIPs) are the two major Zn^2+^ transport families found universally in bacteria, yeasts, plants, and animals, including humans. ZnTs and ZIPs selectively transport Zn^2+^, but in opposite directions: ZnTs export Zn^2+^ from the cytoplasm, whereas ZIPs import Zn^2+^ into the cytoplasm. Thus, ZnTs and ZIPs play important roles in maintaining homeostasis of intracellular and intra-organelle Zn^2+^ levels.

While ZntB from *Escherichia coli* (EcZntB) acts as a Zn^2+^/H^+^ symporter [53], many ZnTs function as proton-driven antiporters, exchanging H^+^ in the extracellular space or organelle lumens for Zn^2+^ in the cytoplasm [41,42,43,44,45,46,47,48,49,50,51,52,53,54,55]. By contrast, there is no clear evidence that ZIPs use proton energy flux to transport Zn^2+^ across the membranes. However, recent biochemical studies suggest that, like ZnTs, *Bordetella bronchiseptica* ZIP (BbZIP) may function as a Zn^2+^/H^+^ antiporter [56].

## 3. ZnTs

ZnTs belong to the cation diffusion facilitator (CDF) family, which can be classified into three groups: Zn-CDFs, Zn/Fe-CDFs, and Mn-CDFs [57,58]. Zn-CDFs consist of Zn^2+^ and Co^2+^ transporters, including ZitB-like, ZnT1-like, and Zrc1-like proteins. The ZitB-like clusters are from *E. coli*. The ZnT1-like clusters include only metazoans. The Zrc1-like cluster includes only fungal CDFs originating from Ascomycetes, Basidiomycetes, and Zygomycetes. Zn/Fe-CDFs are cation-efflux pumps that transport Fe^2+^ or Zn^2+^, and also Co^2+^, Cd^2+^, and Ni^2+^. Mn-CDFs include metal tolerance proteins (MTPs) from plants.

### 3.1. Mammalian ZnTs

Ten ZnTs (ZnTs 1–10) have been identified in mammals, including humans [59,60]. All ZnTs are Zn-CDF members, although ZnT10 is more likely a manganese transporter [59,60,61]. Based on their amino acid sequence similarities, ZnTs are divided into four subgroups: Group 1 includes ZnT5 and ZnT7; group 2 includes ZnT2-ZnT4 and ZnT8; group 3 includes ZnT1 and ZnT10; and group 4 includes ZnT6 and ZnT9 [60]. Most ZnTs form a homodimer composed of the same protomers [50,51,52], whereas ZnT5 and ZnT6 form a heterodimer including two different protomers [62], and all are located on the plasma or organelle membranes, where they control intracellular and extracellular Zn^2+^ balance [59,63]. Specifically, ZnT7 transports Zn^2+^ into the lumen of the pre-*cis*- and *cis*-Golgi, whereas ZnT5/6 and ZnT4 transport Zn^2+^ into the lumen of the *medial*- and *trans*-Golgi [64]. ZnT7 and ZnT5/6 are responsible for the Golgi-to-ER retrograde transport of the ER chaperone ERp44 [64]. This system is involved in the maturation and activation of some secretory proteins during transport through the early secretory pathway [65].

### 3.2. Plant ZnTs

Metal tolerance proteins (MTPs) are bivalent cationic transporters in plants that play crucial roles in metal tolerance and homeostasis in metal non-hyperaccumulators (e.g., *Arabidopsis thaliana*) and hyperaccumulators (e.g., *Arabidopsis halleri* and *Noccaea caerulescens*) [66]. MTPs are classified into seven groups based on their amino acid sequence similarities [67]. Thus, plant MTPs are very diverse so as to satisfy the need to absorb or detoxify specific metals. *A. thalaina* has 12 MTPs, while *P. trichocarpa* MTP has up to 22 MTP genes [68]. In *A. thaliana*, AtMTP1 and AtMTP3 ZnTs localized on the vacuole membrane maintain Zn^2+^ homeostasis [69,70,71]. AtMTP1 and AtMTP3 are involved in the sequestration of excess cytoplasmic Zn^2+^ into vacuoles [71]. Whereas AtMTP1 is more ubiquitously expressed, expression of AtMTP3 is restricted to the root epidermis and cortex [69,72]. Like mammalian ZnT5 and ZnT6, AtMTP5 and AtMTP12 form a heterodimer at the Golgi membrane and transport Zn^2+^ into the Golgi lumen [73].

### 3.3. Yeast ZnTs

Our understanding of ZnTs in yeast derives primarily from *Saccharomyces cerevisiae*. In *S. cerevisiae*, vacuolar ZnTs ZRC1 and COT1 act as Zn^2+^/H^+^ antiporters and regulate Zn^2+^ homeostasis by transporting and storing Zn^2+^ in the vacuole [74,75]. ScZRC1 senses Zn^2+^ availability in the cytosol, possibly through the histidine-repeat motifs, and transports Zn^2+^ from the cytosol to the vacuole when cytosolic Zn^2+^ is abundant, thereby conferring resistance to Zn^2+^ toxicity [76,77].

*S. cerevisiae* also possesses Msc2 and Zrg17, which transport Zn^2+^ from the nucleus and ER to the cytoplasm [78]. ScMsc2 and ScZrg17 interact physically to form a heterodimer and likely serve to maintain the Zn^2+^ levels in the ER of Zn^2+^-adequate cells [79,80,81]. *Schizosaccharomyces pombe* also has a zinc transporter, called ZHF1, which maintains Zn^2+^ homeostasis in the ER and nucleus and sequesters Cd^2+^ into the ER [82]. The structures of yeast ZnTs have not yet been reported. While ScZRC1, ScCOT1, and ScZrg17 are predicted to have six transmembrane (TM) helices, ScMsc2 is presumed to contain up to 16 TM helices.

### 3.4. Bacterial ZnTs

Bacterial ZnTs YiiP, ZitB, and CzcD have been functionally characterized. Insight into the structural features and Zn^2+^ transport mechanisms of bacterial ZnTs comes primarily from YiiP. YiiP was first identified in *Escherichia coli* [83]. In vitro, YiiP also binds Hg^2+^, Co^2+^, Ni^2+^, Mn^2+^, Ca^2+^, and Mg^2+^ but is unlikely to transport them efficiently [84]. Like mammalian ZnTs, YiiP functions as a Zn^2+^/H^+^ antiporter [43,48].

Other ZnTs have been identified recently in bacteria. ZitB conducts Zn^2+^ efflux across the cytoplasmic membrane, thereby reducing Zn^2+^ accumulation in the cytoplasm and rendering bacteria more resistant to Zn^2+^ [85]. By contrast, ZntA, a Zn^2+^-transporting P-ATPase, is required for growth at more toxic concentrations [85]. CzcD is a Cd^2+^, Co^2+^, and Zn^2+^/H^+^-K^+^ antiporter involved in maintaining intracellular divalent cation and potassium homeostasis through active efflux of Zn^2+^, Cd^2+^, and Co^2+^ in exchange for K^+^ and protons [86].

## 4. Structural Basis of ZnTs

In general, ZnTs adopt inward-facing (IF) or outward-facing (OF) forms and undergo conversion between these two forms during Zn^2+^ transport (Figure 1A). The IF form creates an open cavity for Zn^2+^ recruitment from the cytoplasm to the TM metal-binding site. Conversely, in the OF form, an open cavity is formed on the extracellular or organelle luminal side to release Zn^2+^ (Figure 1A). Unlike ZnTs, ZIPs are supposed to operate by a mechanism named elevator-type transport. Membrane transporters using this mechanism commonly consist of moving and fixed domains. Switching between the outward- and inward-facing forms involves the sliding of the entire moving domain through the bilayer as a rigid body. The substrate-binding site translocates some distance across the bilayer during transport, along with the moving domain [87].

The first X-ray crystal structure of ZnTs was solved for EcYiiP in a Zn^2+^-bound OF state [41,42]. Subsequently, cryo-EM single-particle analysis identified the structure of a Zn^2+^-bound IF state for SoYiiP [44,46] (Table 1). More recently, the cryo-EM structures of vertebrate ZnTs human ZnT7 (HsZnT7) [50], human ZnT8 (HsZnT8) [51], and Xenopus ZnT8 (XtZnT8) [52] were reported (Table 1), revealing that these ZnTs can adopt both OF-OF homodimeric and IF-OF heterodimeric conformations [50,51,52]. The presence of the homodimeric and heterodimeric conformations may suggest that the two protomers work independently during Zn^2+^ transport [50,51,52], although more detailed studies are required to draw a firm conclusion.

All Zn-CDFs share common structural folds. The structural core consists of six TM helices (TM1-TM6) in the TM domain (TMD), and a cytosolic domain (CTD) with a ferredoxin-like fold having an αββαβ secondary structure topology (Figure 1B). The Zn^2+^-binding site is formed by an HXXXD motif located on TM2 and TM5, individually, near the center of the TMD (Figure 1C). In some Zn-CDFs, the His (H) and Asp (D) residues on the motifs are replaced by Asp (D) and Asn (N), respectively (Figure 1D). Some eukaryotic Zn-CDFs possess a histidine-rich loop (His-loop) flanked by TM4 and TM5. The length of the His loop, and the number and distribution of His residues in the loop, vary among the Zn-CDFs (Figure 2).

As aforementioned, all ZnTs belong to the SLCA30 family, allowing us to compare the conformational details between them. ZnTs show different TM helix arrangements in both the IF and OF forms. In the IF form of bacterial YiiP and HsZnT8, TM4 and TM5 largely swing to the outside on the cytosolic side (relative to their positions in the OF form), using their luminal ends as pivot points (Figure 3A,B). By contrast, TM2 moves slightly toward the center [51]. Notably, in the IF form of HsZnT7, TM5 kinks at the middle, and its cytosolic half is largely bent toward the outside, resulting in the very open cytosolic cavity (Figure 3C) [50]. In this form, part of the His loop is integrated into the cytosolic cavity, and His164 in this loop is directly coordinated to Zn^2+^ along with His70, Asp74, and Asp244 (Figure 3D) [50]. The detailed mechanistic and functional roles of the His-loop are discussed later.

There are also significant differences in the OF forms of ZnTs with a known structure. The OF form of HsZnT7 has a wider cavity at the luminal side than that of HsZnT8 and EcYiiP due to the more “outside” positions of TM1 and TM2 (Figure 3E,F). An additional conformation has been identified for Zn^2+^-unbound HsZnT7, in which TM5 packs tightly with TM2, TM3, and TM6 on both the cytoplasmic and luminal sides (Figure 3G) [50]. The TM helix arrangement in this state is highly superimposable to that of a previously reported occluded state of Zn^2+^-unbound SoYiiP (Figure 3G) [44]. Although the physiological relevance of the Zn^2+^-unbound occluded state remains unclear, this state may be formed after the release of Zn^2+^ to the luminal or periplasmic side and before complete conversion to the IF state with a widely open cytosolic cavity. Hereafter, we refer to this state as the “IF resting state” in this review article.

## 5. Zn^2+^-Binding Sites on ZnTs

### Zn^2+^-Binding Sites and Metal Ion Selectivity

Metal specificity is an important functional feature of all metal transporters. X-ray crystallographic and cryo-EM analyses revealed one to three Zn^2+^-binding sites (site A, site B, and site C) in ZnTs. All ZnTs commonly possess a Zn^2+^-binding site in the TMD (site A). Zn^2+^ binds transiently to site A before effluxing to the other side, indicating that site A is located on the Zn^2+^-translocation pathway. Site A contains highly conserved Zn^2+^-binding motifs formed by the BXXXB motif on TM2 and TM5, in which B is His (H), Asp (D), Asn (N), or Glu (E), and X can be any residues (Figure 1C,D). There are, however, some differences in the sequence of site A (Figure 1D). While human, plant, and yeast ZnTs conserve the (HXXXD)^TM2^-(HXXXD)^TM5^ motif, different amino acid sequences are seen in the motifs of bacterial ZnTs; for instance, (DXXXD)^TM2^-(HXXXD)^TM5^ in EcYiiP and SoYiiP, and (HXXXD)^TM2^-(HXXXD)^TM5^ in EcZitB and CzcD (Figure 1D). Notably, mammalian ZnT6 loses the BXXXB motif in the TMD, and does not therefore have Zn^2+^ transport activity [62]. Mammalian ZnT10 harboring (NXXXD)^TM2^-(HXXXD)^TM5^ transports Mn^2+^ rather than Zn^2+^ (Figure 1D) [88,89,90]. AtMTP11 and ShMTP8 harbor (DXXXD)^TM2^-(DXXXN)^TM5^ and transport both Mn^2+^ and Cu^2+^ (Figure 1D) [66]. Thus, not all ZnTs transport Zn^2+^ exclusively.

Site B is less conserved at the TMD-CTD interface (Figure 1B and Figure 4A). In bacterial YiiP, this site is located in the TM2-TM3 loop, which contains a DHH motif (Figure 4B) [42,43,44]. In HsZnT8, this site is constituted by His residues from the TM2-TM3 loop in the TMD and from the α2-β3 loop in the CTD, and it has low affinity for Zn^2+^ [51]. While the functional role of site B is unclear in HsZnT8, bacterial YiiP participates directly in Zn^2+^ transport [43,44]. Since site B is positioned near the cytosolic Zn^2+^ entry gate, mutation of two His residues at site B reduces the Zn^2+^ transport activity of bacterial YiiP [43]; thus, site B is thought to trap Zn^2+^ in the cytosol, thereby facilitating Zn^2+^ transport [43,51]. Although the structure of AtMTP1 has not been solved, homology modeling based on the crystal structure of EcYiiP suggests that it lacks site B (Figure 4B). However, mutations of some residues in the TM2-TM3 loop impair the Zn^2+^ transport activity of AtMTP1, indicating that the TM2-TM3 loop itself is essential for AtMTP1 function [91].

Site C is located at the dimer interface between two CTDs (Figure 1B and Figure 4A). In bacterial YiiP, site C forms a binuclear Zn^2+^ complex composed of the (HHD)_2_ motif and stabilizes its dimeric conformation (Figure 4C,D) [42,43]. In HsZnT8, the exact location of site C differs from that in bacterial YiiP. In HsZnT8, two Zn^2+^ ions are coordinated by an HCH motif from the N-terminal loop, and a Cys-Cys pair from the C-terminal tail, thereby forming a tetrahedral complex (Figure 4D) [51]. The HCH motifs seal off site C and bury the Zn^2+^ ions inside the protein (Figure 4D). This motif is highly conserved among the ZnT8 orthologues but not among bacterial ZnTs [51]. The N-terminal truncation that accompanies the loss of the HCH motif reduces the Zn^2+^ uptake activity of HsZnT8 significantly [51]. AtMTP1 also possesses site C within the CTD (Figure 4C), and the lack of this domain results in loss of function [91]. Thus, Zn^2+^-mediated dimerization via the CTD seems likely to be essential for Zn^2+^ transport by HsZnT8, bacterial YiiP, and plant MTP1.

By contrast, no Zn^2+^ ions have been identified at either site B or site C of HsZnT7, although its cryo-EM structures were determined in the presence of high Zn^2+^ concentrations (10, 200, or 300 μM) of Zn^2+^ (Figure 4). Consistent with this, residues required for Zn^2+^ binding are not conserved at site B or site C of HsZnT7 (Figure 4C). Presumably, other elements contribute to the dimerization of the CTD and the Zn^2+^ transport activity of HsZnT7. Indeed, dimerization of the CTD is mediated by residues within four β-strands and TM2-TM3 loops located at the dimer interface (Figure 4A) [50].

## 6. Mechanism of Zn^2+^ Transport by Human ZnT7 and Bacterial YiiP

### 6.1. Zn^2+^ Transport by HsZnT7

HsZnT7 transports Zn^2+^ from the cytoplasm to the TM Zn^2+^-binding site (site A), and then to the Golgi lumen. The cryo-EM structures of HsZnT7 in multiple states have helped to paint a full picture of Zn^2+^ transport mediated by this transporter (Figure 5) [50]. In the absence of Zn^2+^, TM5 packs against TM2, TM3, and TM6 at both the cytosolic and luminal sides, forming an “IF resting” state (Figure 3G and Figure 5A). In this state, Zn^2+^ uptake to site A seems to be blocked due to the closed Zn^2+^ entry gate (Figure 5A–C(i)). In the presence of Zn^2+^, however, the cytosolic cavity opens by bending the N-terminal half of TM5, and the His-loop is integrated into the cavity to coordinate with Zn^2+^ at site A (Figure 5A–C(ii)). In the next step, TM5 returns to a straight conformation concomitant with pulling His164 out of site A. Consequently, His240 on TM5 coordinates with Zn^2+^ instead of His164 (Figure 5A–C(iii)). Upon conversion to the OF state, the His70 side chain moves away from Zn^2+^ slightly, while Asp74, His240, and Asp244 remain coordinated with Zn^2+^ (Figure 5B(iv)). Simultaneously (or shortly thereafter), protons presumably enter the Zn^2+^ transport channel from the weakly acidic Golgi lumen to protonate His70 and His240. As a result, His70 flips out of site A, and Zn^2+^ is released to the Golgi lumen (Figure 5A–C(v)). In this context, it is notable that while a His residue is highly conserved among human ZnTs, it is substituted by an Asp residue in bacterial YiiP (Figure 1C,D). Although it is not fully elucidated how many protons are involved in the release of Zn^2+^, the above mechanistic model suggests that HsZnT7 exchanges one Zn^2+^ for two protons per cycle.

### 6.2. Zn^2+^ Transport by Bacterial YiiP

Recent biochemical, structural, and computational studies provide deep insight into the mechanism of Zn^2+^ transport mediated by bacterial YiiP [43,44]. Initially, Zn^2+^ is recruited to site B, formed by the TM2-TM3 loop, and then transferred to site A of another protomer (Figure 6) [43,44]. In the Zn^2+^-free IF state, the TM2-TM3 loop is disordered to allow Zn^2+^ to approach site B. After binding Zn^2+^ site B, the TM2-M3 loop forms an ordered conformation [43]. Importantly, the higher affinity of Zn^2+^ for site A than site B explains sequential Zn^2+^ transfer from site A to site B [43]. At this step, the negatively charged cytoplasmic cavity seems advantageous for directing Zn^2+^ to site A [43,92]. In the OF state, the TM2-TM3 loop in a Zn^2+^-free state interacts with the CTD, forming an extended conformation [43]. Zn^2+^ is released to the periplasm and is facilitated by the protonation of two residues in site A, or potentially three residues at a lower pH [43].

MD simulations using EcYiiP and SoYiiP as a model metal transporter demonstrate that Zn^2+^ binds transiently to Asp150 (Asp152 in SoYiiP) on TM5, and to Glu79 (Glu81 in SoYiiP) on TM3, before reaching site A (Figure 6) [92]. In the next step, His153 (His155 in SoYiiP), located at site A, plays a key role in guiding Zn^2+^ to site A. In support of this, His153 can adopt two conformations to receive Zn^2+^ from Asp150, after which it forms a tetrahedral Zn^2+^ complex with Asp45, Asp49, and Asp157 (Figure 6) [49]. Release of Zn^2+^ can be triggered by protonation of site A residues [43,44]. His153 is likely deprotonated again when it faces the neutral-pH cytoplasm to receive Zn^2+^ from site B via Asp150 [49,51,54,92,93].

### 6.3. Role of the Proton Gradient in ZnT-Mediated Zn^2+^ Transport

Lactose permease (LacY) utilizes a proton motive force for the efficient transport of lactose [94]. Presumably, ZnTs also uses the proton motive force to transport Zn^2+^ from the cytoplasm to the extracellular space or the organelle lumen [55]. Given that the concentration of labile Zn^2+^ in the Golgi is much higher than in the cytosol [64,95,96], and that the Golgi is a weakly acidic organelle [97], it makes sense that Golgi-resident ZnTs utilize the energy provided by the Golgi-to-cytosol proton flow to transport Zn^2+^ to the luminal side. As mentioned above, protonation of His70 and His240 is critical for the IF-to-OF conversion and subsequent release of Zn^2+^. While HsZnT7 exchanges one Zn^2+^ for (possibly) two H^+^ per cycle, EcZitB [47] and EcYiiP [48] exchange one Zn^2+^ for one H^+^. SoYiiP conducts the exchange at a ratio of 1:2–3, depending on the external pH [43]. Human ZnT2 is thought to operate at a ratio of 1:2 [54].

## 7. Structural and Functional Roles of the Histidine-Rich Loop (His-Loop)

The His-loop is a unique sequence located at the cytoplasmic side between TM4 and TM5 (Figure 1B and Figure 2). The length of this loop, as well as the number and arrangement of His residues, can differ (Figure 2). Most bacterial ZnTs (e.g., YiiP and MntE), several mammalian ZnTs (e.g., ZnTs 6, 9, and 10), and plant MTPs (e.g., AtMTPs 5, 6, and 11) possess a short TM4-TM5 loop without any His residues. Mammalian ZnTs such as ZnTs 2-4 and ZnT8 are also characterized by a short His-loop. Of note, several mammalian ZnTs (e.g., ZnTs 1, 5, and 7), plant MTP1s (e.g., AtMTP1, NgMTP1, OsMTP1, and HvMTP1), and yeast ZnTs (e.g., SpZHF1, ScZRC1, and ScCOT1) have an exceptionally long His-loop (Figure 2). Despite having a relatively long TM4-TM5 loop, ZnT10 has no His residues in the loop (Figure 2). Such variations in the His-loop may underlie the diverse functions and different mechanisms of Zn^2+^ transport mediated by ZnTs. Indeed, the His-loop also functions as a Zn^2+^-buffering pocket [98], a Zn^2+^ selectivity filter [99], a Zn^2+^ sensor [98,100,101], or a Zn^2+^-fishing loop [50].

As described above, cryo-EM analysis of HsZnT7 revealed that a short segment of the His-loop inserts into the cytoplasmic cavity, where His164 contained in the His-loop coordinates directly with Zn^2+^ at site A, while His240 on TM5 is kicked out from site A due to bending of the N-terminal half of TM5 (Figure 5A,B(ii)) [50]. Thus, the His-loop appears to contribute to efficient Zn^2+^ uptake to site A, but its task seems transient. In other words, HsZnT7 may use the His-loop as a “Zn^2+^ fishing loop”, and His residues in the loop function as a “fishing hook” that captures Zn^2+^ and transfers it to site A. Consistent with this, the deletion of the His-loop, or the mutations of all His residues in the loop to Ser, abolishes the Zn^2+^ transport activity of HsZnT7 [50]. Similarly, deletion of the entire His-loop segment in *Noccaea caerulescens* MTP1 and AtMTP1 results in substantial loss of Zn^2+^ transport activity [98,102] although deletion of the first half of the His-loop increases the activity of AtMTP1 significantly [98].

The 63-residue His-loop of HsZnT7 contains 21 His residues and can be divided into two halves: The first half contains seven His residues close to the C-terminus of TM4, and the second half contains 14 His residues close to the N-terminus of TM5 (Figure 2) [50]. The His-loop of HsZnT7 binds Zn^2+^ with a K_d_ value of 12 μM; although each half can bind one Zn^2+^, the first and second halves appear to bind Zn^2+^ co-operatively [50]. AtMTP1 also contains an exceptionally long His-loop containing 25 His residues and can also be divided into two halves: The first half contains 18 His residues close to the C-terminus of TM4, while the second half contains seven His residues close to the N-terminus of TM5 (Figure 2) [98]. The His-loop of AtMTP1 can bind four Zn^2+^ ions, with a K_d_ value of 25 μM [101]. The lack of either half leads to impairment of Zn^2+^-binding by AtMTP1, suggesting that the two halves of the His-loop work co-operatively to form multiple Zn^2+^-binding sites [101]. However, stable Zn^2+^ binding may lead to slower Zn^2+^ release from AtMTP1, which likely explains the less efficient Zn^2+^ transport by AtMTP1 WT than by the mutant lacking the first half [98,101]. Consistent with this, the His-loop of AtMTP1 adopts a β-strand conformation to suppress its Zn^2+^ transport activity at high concentrations of Zn^2+^, implying a role in sensing excessive Zn^2+^ levels in the cytosol [101].

## 8. Zn^2+^ Transport by Other Zn^2+^ Transport Systems

### 8.1. ZIPs

ZIPs play a crucial role in mediating metal import across the plasma membrane as well as export across organelle membranes. ZIPs are classified into four subfamilies based on phylogenetic relationships [102]. In humans, 14 ZIPs have been identified. Despite their relevance to human diseases, the structural and mechanistic details of mammalian ZIPs remain poorly understood. Recent studies on *Bordetella bronchiseptica* ZIP (BbZIP) have shed light on some structural and mechanistic features [56,103,104]. Crystal structures of BbZIP revealed an IF monomer, with binuclear metal-binding sites potentially occupied by Zn^2+^ or Cd^2+^ [104], while cryo-EM studies revealed an IF homodimer [56]. BbZIP is an elevator-type transporter in which a four-TM helix bundle with bound metal ions slides as a rigid body against the dimeric domain composed of the other static TM helices to exert an alternating access mechanism, and its Zn^2+^ uptake is pH-dependent [56,105,106,107]. However, it is unclear whether ZIPs transfer Zn^2+^ through a symport or antiport mechanism [56,105]. The presence of a His-rich loop in many mammalian ZIPs, including BbZIP, adds complexity to the mechanisms underlying their regulated Zn^2+^ transport [108,109]. Plants possess more ZIPs [110], but detailed structural and mechanistic characterization has not been reported for any of them. 

### 8.2. ZntB

ZntB belongs to the prokaryotic CorA family [111,112]. *E. coli* and *P. aeruginosa* ZntB transport multiple metal ions, including Zn^2+^, Co^2+^, Ni^2+^, and Cd^2+^ [53,113]. ZntB forms homogeneous pentagonal structures that span the biological membrane, similar to other members of the CorA family. Each protomer consists of two TM helices and a large cytoplasmic domain [53,113]. ZnTB-mediated Zn^2+^ transport is stimulated by a pH gradient across the biological membrane. ZntB transports protons and Zn^2+^ together in the same direction, thereby working as a symporter [53]. In contrast to bacterial YiiP and human ZnTs, ZntB does not seem to use an alternating access mechanism mediated by conformational transitions between IF and OF forms [53]. The cryo-EM structure of PaZntB is similar to that of EcZntB and *T. maritima* CorA, but with different helical arrangements [53,113,114]. PaZntB is capable of transporting Zn^2+^, Mg^2+^, Cd^2+^, Ni^2+^, and Co^2+^. In contrast to other ZntBs, PaZntB is unlikely to use the proton motive force [53,113]. Thus, different mechanisms may operate among ZntBs from various bacterial species [53,111,113,115].

### 8.3. P-Type ATPases with Zn^2+^ Transport Activity

Zn^2+^-transporting P-type ATPases, which belong to class IB (P_IB-2_-ATPases), are active zinc transporters. While no P-type ATPases with Zn^2+^ transport activity have been identified in mammals, several have been identified in bacteria and plants. In plants, the P_IB-2_-ATPases, also known as the heavy metal ATPases (HMAs), are thought to play an important role in the transport of transition metals. In *A. thaliana*, AtHMA2 and AtHMA4 are zinc transporters that increase Zn^2+^ levels in roots, stems, and leaves [66,116]. AtHMA1 functions to detoxify Zn^2+^ in the chloroplast, while AtHMA3 is thought to supply Zn^2+^ to the ER and vacuole. Little is known about the structural features and Zn^2+^ transport mechanisms of HMAs [66]. In bacteria, ZntA is an active zinc transporter that is crucial for cellular detoxification and sub-cellular redistribution of Zn^2+^ [117]. ZntA couples ATP hydrolysis with Zn^2+^ transport via the “Post-Albers” cycle, during which at least four primary intermediate states (E1, E1P, E2P, and E2) accumulate [118,119,120,121]. Structurally, P_IB-2_-ATPases possess four domains: A TMD, a nucleotide-binding domain (N domain), an actuator/dephosphorylation domain (A domain), and a phosphorylation domain (P domain) [118,120,122]. In addition, P_IB-2_-ATPases contain metal-binding domains (MDBs) in the N-terminal region to promote metal recruitment to the TM metal-binding site (114, 116, 118). To date, only the crystal structure of *Shigella sonnei* ZntA has been reported [117], and the cryo-EM structures of other P_IB-2_-ATPases are currently unknown.

### 8.4. Zn^2+^ Transport by ATP-Binding Cassette Transporters

ATP-binding cassette (ABC) transporters are a ubiquitous superfamily of integrated membrane proteins that transport various substrates across biological membranes by utilizing ATP as an external energy source [123]. In bacteria, ZnuABC is a specialist zinc transporter. It comprises three protein subunits: A periplasmic Zn^2+^-binding protein (ZnuA), an integrated membrane protein that transports Zn^2+^ across the cytoplasmic membrane (ZnuB), and an ATPase protein in charge of coupling Zn^2+^ transport to ATP hydrolysis (ZnuC) [124,125]. In most Gram-negative bacteria, ZnuABC expression is regulated by a Zur (zinc uptake regulatory) protein induced by Zn^2+^ acquisition [126]. The structure of ZnuABC and its mechanism of Zn^2+^ transport coupled to ATP hydrolysis are only poorly understood, although crystal structures have been solved for EcZnuA [124,127,128], and *Salmonella enterica* ZnuA [129].

## 9. Conclusions

As described above, while some members of the Zn-CDF family share structural and mechanistic features, there are considerable variations among them. It is widely accepted that ZnTs commonly operate as dimers; however, detailed structural analyses conducted so far show that whereas bacterial YiiPs adopt OF-OF and IF-IF homodimers, human ZnTs exist as either OF-OF homodimers or IF-OF heterodimers. No IF-IF homodimers have been observed for mammalian ZnTs, likely due to their conformational instability. Indeed, modes of the TM helix rearrangement between the IF and OF states among ZnTs seem to differ significantly. Of particular note, HsZnT7 undergoes marked bending of TM5 to allow efficient Zn^2+^ uptake. Concomitantly, a part of the long cytoplasmic His-loop is integrated into the negatively charged cytosolic cavity to facilitate acqusition and efficient transfer of Zn^2+^ to the TM Zn^2+^-binding site. In this state, His164 in the His-loop is coordinated directly with Zn^2+^ in the TMD. Subsequently, His240 (TM5) repaces His164, followed by the conversion from the IF to the OF state. In this context, the essential role of His164 needs to be further explored by conducting additional mutational and structural studies. It will also be interesting to see if another His residue in the His-loop can replace His164 for efficient Zn^2+^ recruitment when His164 is deleted. Recently, a de novo heterozygous variant of *SLC30A7*, His164Ser, was found in Joubert syndrome patients [130]. Although no *SLC30A7* variants have yet been shown to cause human phenotypes or diseases, *SLC30A7* is a candidate gene associated with Joubert syndrome [130].

Undoubtedly, multiple ZnTs work co-operatively to maintain Zn^2+^ homeostasis in cells. Indeed, disruption of Zn^2+^ homeostasis leads to disruption of protein homeostasis [64,131] and dysfunction of essential enzymes, transcription factors, and other biomolecules [132,133,134,135], eventually causing many fatal diseases [29,30,31,32,33,34,35,36,37,38,39,40]. In this regard, the mechanisms of Zn^2+^ homeostasis mediated by various kinds of ZnTs need to be understood more comprehensively. Zinc biology is an important field of research that encompasses biochemical, structural, computational, physiological, and medical approaches.

## Figures and Tables

**Figure 1 ijms-25-03045-f001:**
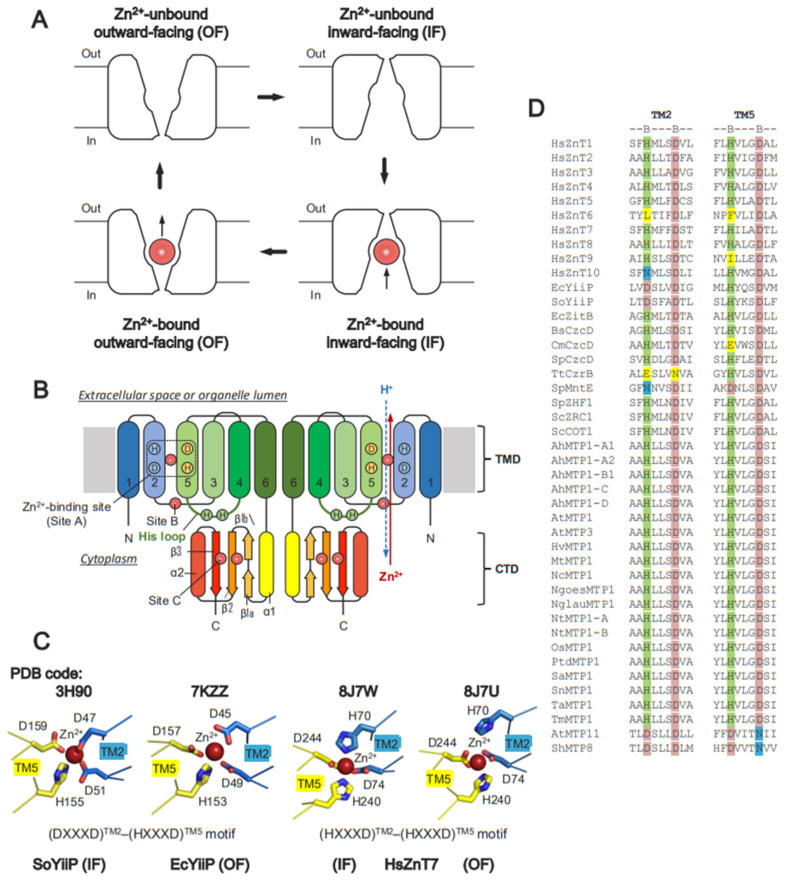
Molecular architecture of zinc transporters (ZnTs). (**A**) Two conformations of ZnTs, inward-facing (IF) and outward-facing (OF), with the metal-binding site open to the cytoplasmic side (In) and extracellular or organelle luminal side (Out), respectively. ZnTs convert between these two forms during the Zn^2+^ transport cycle. (**B**) Typical topology model of a ZnT homodimer in which two transmembrane domains (TMDs) and two C-terminal domains (CTDs) tightly contact each other. The Zn^2+^-binding sites are labeled as site A in the TMD, site B at the interface of the TMD and CTD, and site C in the CTD. Red and blue arrows indicate the directions of Zn^2+^ and H^+^ transports, respectively. For clarity, the arrows are shown only in a right protomer. (**C**) Zn^2+^ coordination structures of bacterial YiiP (PDB code: 3H90 for SoYiiP in the IF form and 7KZZ for EcYiiP in the OF form) and human ZnT7 (PDB code: 8J7W for the IF form and 8J7U for OF form). Red spheres indicate bound Zn^2+^. “A” in the red sphere indicates Zn^2+^ bound to site A. (**D**) Sequence alignment of Zn^2+^ coordinating residues at site A. Light green and light pink highlight conserved His and Asp residues, respectively. Yellow highlights loss of the conserved His and Asp residues. Cyan highlights conserved Asn residues required for Mn^2+^ transport.

**Figure 2 ijms-25-03045-f002:**
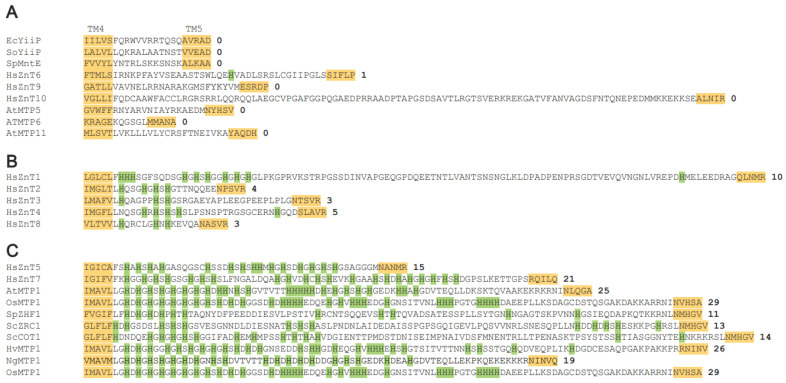
Amino acid sequence of the histidine-rich loop (His-loop) of ZnTs. Some ZnTs have no His-loop (**A**), while others have a short (**B**) or exceptionally long His-loops (**C**). Histidine residues are highlighted in light green. The C-terminal part of TM4 and the N-terminal part of TM5 are shown in orange. Bold numbers on the right denote the number of histidine residues contained in the His-loop. Accession numbers are as follows: *Escherichia coli* YiiP (EcYiiP, UniProt code: P69380), *Shewanella oneidensis* YiiP (SoYiiP, UniProt code: Q8E919), *Streptococcus pneumoniae* MntE (SpMntE, UniProt code: Q8DP19S), *Homo sapiens* ZnT6 (HsZnT6, UniProt code: Q6NXT4), *Homo sapiens* ZnT9 (HsZnT9, UniProt code: Q6PML9), *Homo sapiens* ZnT10 (HsZnT10, UniProt code: Q6XR72), *Arabidopsis thaliana* MTP5 (AtMTP5, UniProt code: Q6ICY4), *Arabidopsis thaliana* MTP6 (AtMTP6, UniProt code: Q8L725), *Arabidopsis thaliana* MTP11 (AtMTP11, UniProt code: O80632), *Homo sapiens* ZnT1 (HsZnT1, UniProt code: Q9Y6M5), *Homo sapiens* ZnT2 (HsZnT2, UniProt code: Q9BRI3), *Homo sapiens* ZnT3 (HsZnT3, UniProt code: Q99726), *Homo sapiens* ZnT4 (HsZnT4, UniProt code: O14863), *Homo sapiens* ZnT8 (HsZnT8, UniProt code: Q8IWU4), *Homo sapiens* ZnT5 (HsZnT5, UniProt code: Q8TAD4), *Homo sapiens* ZnT7 (HsZnT7, UniProt code: Q8NEW0), *Arabidopsis thaliana* MTP1 (AtMTP1, UniProt code: Q9ZT63), *Oryza sativa* MTP1 (OsMTP1, UniProt code: Q688R1), *Schizosaccharomyces pombe* ZHF1 (SpZHF1, UniProt code: O13918), *Saccharomyces cerevisiae* ZRC1 (ScZRC1, UniProt code: P20107), *Saccharomyces cerevisiae* COT1 (ScCOT1, UniProt code: P32798), *Hordeum vulgare* MTP1 (HvMTP1, UniProt code: A0JJL9), and *Noccaea goesingensis* MTP1 (NgMTP1, UniProt code: Q6Q4F7).

**Figure 3 ijms-25-03045-f003:**
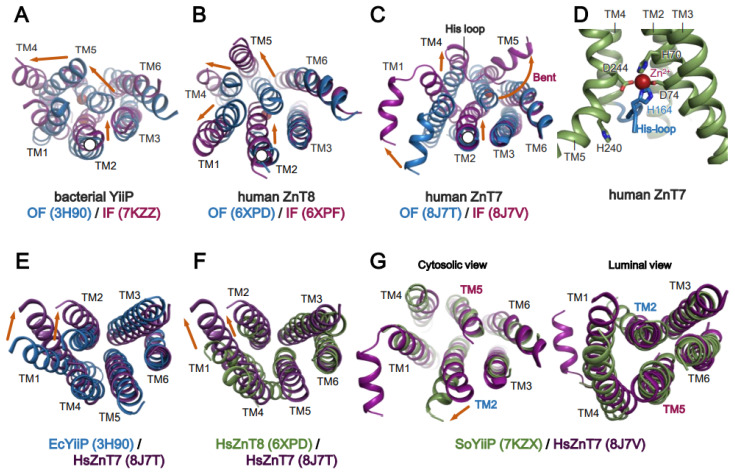
Rearrangement of the transmembrane helix of ZnTs during transition between the OF and IF forms. (**A**–**C**) Superimpositions of the OF (blue) and IF (magenta) forms of bacterial YiiP (PDB codes: 3H90 and 7KZZ, respectively), human ZnT8 (PDB codes: 6XPD and 6XPF, respectively), and human ZnT7 (PDB codes: 8J7T and 8J7V, respectively), viewed from the cytosolic side. (**D**) Integration of the His-loop of HsZnT7 into the widely open cytosolic cavity, viewed from the side. (**E**) Superimposition of the OF forms of EcYiiP (blue, PDB code: 3H90) and HsZnT7 (violet, PDB code: 8J7T), viewed from the luminal side. (**F**) Superimposition of the OF forms of HsZnT8 (green, PDB code: 6XPD) and HsZnT7 (violet, PDB code: 8J7T), viewed from the luminal side. (**G**) Superimposition of the Zn^2+^-unbound IF states of HsZnT7 (violet, PDB code: 8J7V) and SoYiiP (green, PDB code: 7KZX), viewed from the cytosolic side. The cytosolic domains and TM helix loops are omitted for clarity. Orange arrows indicate the movement of TM helices during the conversion from the OF to the IF state. TM, transmembrane.

**Figure 4 ijms-25-03045-f004:**
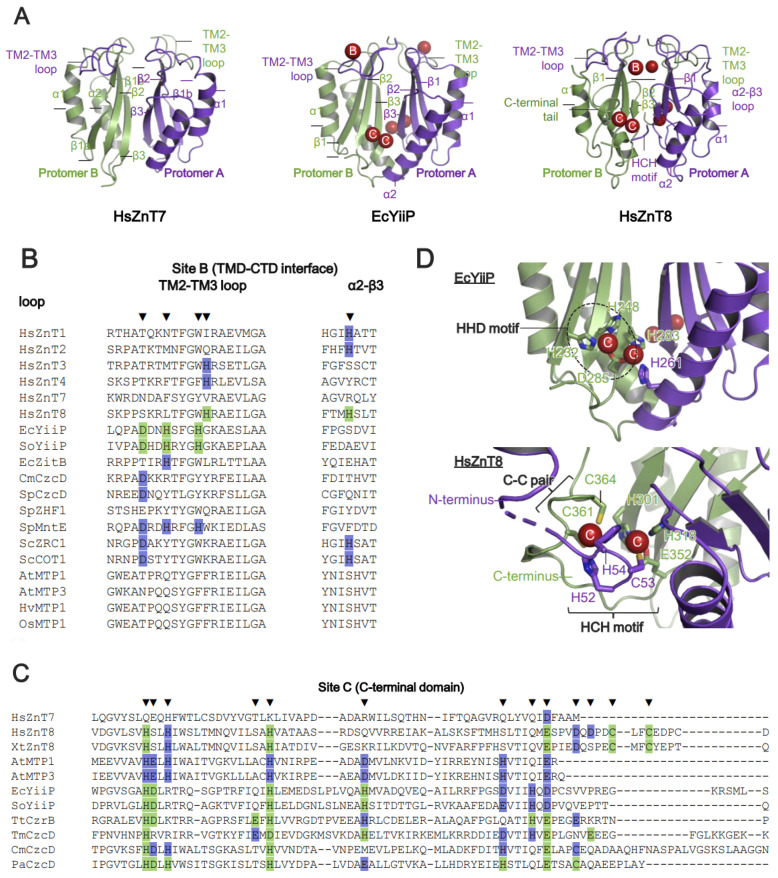
Zn^2+^-binding sites in the C-terminal domain of ZnTs. (**A**) The C-terminal domain (CTD) of dimeric HsZnT7 (**left**), EcYiiP (**middle**), and HsZnT8 (**right**). Red spheres indicate Zn^2+^ ions. “B” and “C” in the red spheres indicate Zn^2+^ bound to site B and site C, respectively. Note that HsZnT7 contains no Zn^2+^ binding sites in the CTD. (**B**) Sequence alignment of representative ZnT-family members from human, bacteria, and plants around site B, located at the TMD-CTD interface. Zn^2+^-coordinating residues confirmed by structure analysis are highlighted in light green and marked by black down-pointing triangles. Light violet highlights His and Asp residues that are predicted to be coordinated with Zn^2+^. (**C**) Sequence alignment of representative ZnT-family members around site C in the CTD. Zn^2+^-coordinating residues identified by structure analyses are highlighted in light green and marked by black down-pointing triangles. Light violet highlights residues to be involved in Zn^2+^ coordination. Note that HsZnT7 lacks site B and site C. TM, transmembrane. (**D**) Closed-up views of Zn^2+^-binding sites around the CTD (site C) in EcYiiP (upper) and HsZnT8 (lower). Bound Zn^2+^ ions are represented by red spheres. Black circle indicates the HHD motif located at site C of EcYiiP. “C” in the red spheres indicate Zn^2+^ bound to site C. Accession numbers are as follows: *Homo sapiens* ZnT1 (HsZnT1, UniProt code: Q9Y6M5), *Homo sapiens* ZnT2 (HsZnT2, UniProt code: Q9BRI3), *Homo sapiens* ZnT3 (HsZnT3, UniProt code: Q99726), *Homo sapiens* ZnT4 (HsZnT4, UniProt code: O14863), *Homo sapiens* ZnT7 (HsZnT7, UniProt code: Q8NEW0), *Homo sapiens* ZnT8 (HsZnT8, UniProt code: Q8IWU4), *Escherichia coli* YiiP (EcYiiP, UniProt code: P69380), *Shewanella oneidensis* YiiP (SoYiiP, UniProt code: Q8E919), *Escherichia coli* ZitB (EcZitB, UniProt code: P75757), *Cupriavidus metallidurans* CzcD (CmCzcD, UniProt code: P13512), *Streptococcus pneumoniae* CzcD (SpCzcD, UniProt code: A0A0B7LW62), *Schizosaccharomyces pombe* ZHF1 (SpZHF1, UniProt code: O13918), *Streptococcus pneumoniae* MntE (SpMntE, UniProt code: Q8DP19S), *Saccharomyces cerevisiae* ZRC1 (ScZRC1, UniProt code: P20107), *Saccharomyces cerevisiae* COT1 (ScCOT1, UniProt code: P32798), *Arabidopsis thaliana* MTP1 (AtMTP1, UniProt code: Q9ZT63), *Arabidopsis thaliana* MTP3 (AtMTP3, UniProt code: Q9LXS1), *Hordeum vulgare* MTP1 (HvMTP1, UniProt code: A0JJL9), *Oryza sativa* MTP1 (OsMTP1, UniProt code: Q688R1), *Thermus thermophilus* CzrB (TtCzrB, UniProt code: Q8VLX7), *Thermotoga maritima CzcD* (TmCzcD, UniProt code: Q9WZX9), and *Pseudomonas aeruginosa CzcD* (PaCzcD, UniProt code: Q9I6A3).

**Figure 5 ijms-25-03045-f005:**
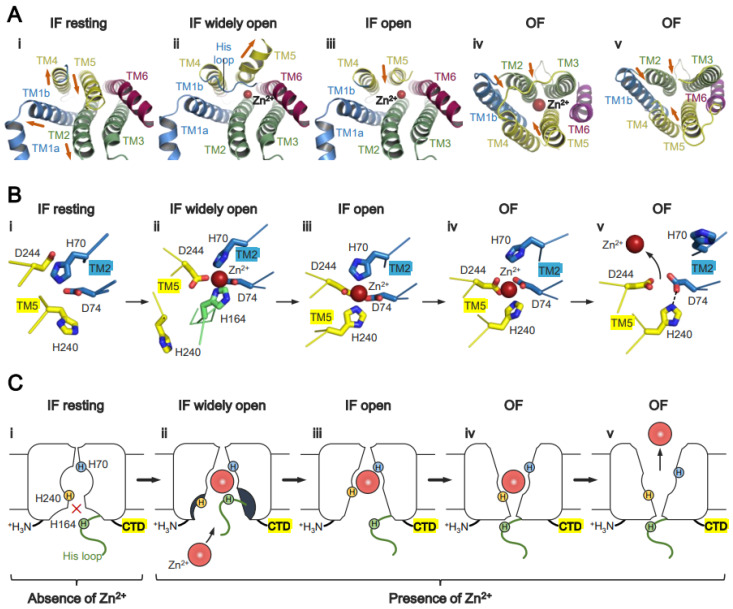
Zn^2+^ transport mechanism used by human ZnT7. (**A**) TM helix rearrangement during the transition from the IF state (**i**–**iii**) to the OF state (**iv**,**v**), viewed from the cytosolic side (**i**–**iii**) and from the luminal side (**iv**,**v**). (**B**) Zn^2+^-coordination structure at site A. Note that His240 (TM5) and His70 (TM2) undergo striking positional shifts during the Zn^2+^ transport cycle. His164 (His-loop) is coordinated directly with Zn^2+^ in the “IF widely open” state and is subsequently replaced by His240 (TM5). His70 moves away from site A to facilitate Zn^2+^ release to the luminal side. (**C**) Simplified cartoon showing the Zn^2+^ transport cycle of human ZnT7. States **i**–**v** indicate intermediates generated during the transition from the IF state (**i**–**iii**) to OF states (**iv**,**v**).

**Figure 6 ijms-25-03045-f006:**
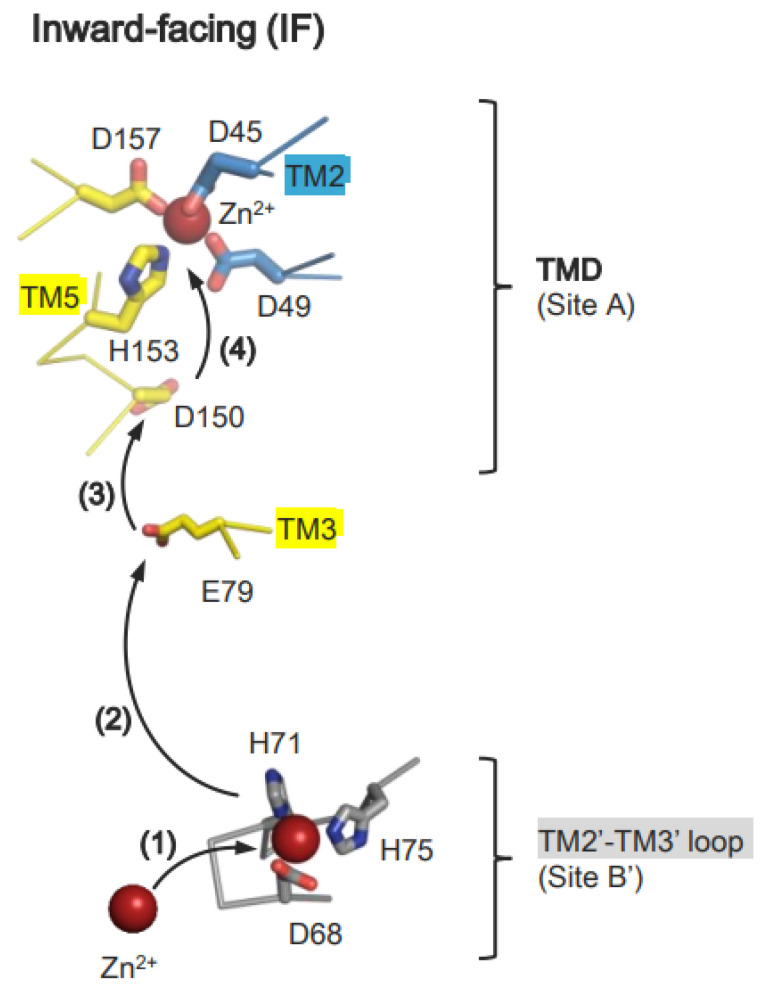
Mechanism of Zn^2+^ recruitment and transfer to site A by bacterial YiiP. Initially, Zn^2+^ is recruited to site B (step (**1**)) located at the TMD-CTD interface, and then transferred to site A via steps (**2**–**4**), sequentially.

**Table 1 ijms-25-03045-t001:** X-ray and cryo-EM structure of zinc transporters (ZnTs).

Proteins	Main Functions	Organisms	States	Conformations (PDB Code)	Ligands	Methods	References
YiiP	Transport Zn^2+^ out of the cytoplasm and into the periplasm	*Escherichia coli*	Homodimer	Outward-facing (2QFI, 3H90)	Zn^2+^	X-ray diffraction	[41,42]
		*Shewanella oneidensis*	Homodimer	Inward-facing (3J1Z, 5VRF, 7KZZ ^(1)^)	Zn^2+^	Electron microscopy	[44,45,46]
			Homodimer	Inward-facing occluded (7KZX)	Zn^2+^		[43]
ZnT7	Transport Zn^2+^ out of the cytoplasm and into the Golgi lumen	*Homo sapiens*	Homodimer	Outward-facing (8J7T)	Apo	Electron microscopy	[50]
			Homodimer	Outward-facing (8J7U)	Zn^2+^		
			Heterodimer	Inward-facing and outward-facing (8J7V ^(2)^)	Apo		
			Heterodimer	Inward-facing with Zn^2+^ and outward-facing (8J80 ^(3)^)	Zn^2+^, Apo		
			Heterodimer	Inward-facing with Zn^2+^ and outward-facing with Zn^2+^ (8J7W) ^(4)^	Zn^2+^		
ZnT8	Transport Zn^2+^ out of the cytoplasm and into the insulin secretory granule	*H. sapiens*	Homodimer	Outward-facing (6XPE)	Zn^2+^	Electron microscopy	[51]
			Heterodimer	Outward-facing and inward-facing (6XPF)	Apo		
		*Xenopus tropicalis*	Homodimer	Outward-facing (7Y5G)	Zn^2+^		[52]
			Homodimer	Outward-facing (7Y5H ^(5)^)	Apo		

^(1)^ This structure was observed in the presence of 0.5 mM EDTA. ^(2)^ This structure was observed in the absence of Zn^2+^. ^(3)^ This structure was observed in the presence of 10 μM Zn^2+^. ^(4)^ This structure was observed with addition of 200 and 300 μM Zn^2+^. ^(5)^ This structure was observed at low pH.

## Data Availability

Not applicable.

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
