# Peer review of "Structures, Mechanisms, and Physiological Functions of Zinc Transporters in Different Biological Kingdoms"

_ijms, 2024, doi:10.3390/ijms25053045_

Round 1
Reviewer 1 Report
Comments and Suggestions for Authors
This manuscript represents an overview of the structure and function
of known Zinc ion transporter proteins. I am not an expert in this
field, so I cannot assess how comprehensive a review it is, but
certainly many classes of transporters are covered, occurring in
bacteria, plants and animals. Two classes of Zinc transporter are
described in some mechanistic detail, in terms of structure and
molecular rearrangements which occur during the transfer of a zinc ion
across a membrane. The manuscript is well written.
While reading, One question arose is about mechanistic differences for
inward versus outward ion flow. The two classes of transporter
described in detail appear to pump metal ions from inside the
cell/organelle to the outside. The authors might comment on how
retrograde mechanisms may differ.
A few typos/misprints were noted in reading:
1) Fig. 5 is placed before Fig. 4 in the version I read.
2) First sentence of the Introduction: singular/plural are mixed and
there is an extra ``of''
3) p. 1, line 33: ``In addition, ... also'' : only one should be
present.
4) p. 5. line 165: ``make tightly contact'' should be fixed.
5) p. 14, line 479 ``work co-operatively work'' should be fixed.
details in comment above
Reviewer 2 Report
Comments and Suggestions for Authors
The review discusses zinc transporters in different organisms.
I recommend respecting the same order in the title (structure, mechanisms and physiological functions) and in the text.
I suggest also highlighting the 2 types of transporters, those who export and those who import, in the titles of the paragraphs.
In the introduction I suggest discussing that excess zinc can be harmful.
In the text the authors discuss the ZnT transporter first in mammals, then in plants, then in yeast and finally in bacteria: it does not respect the order of the title. I believe it is more correct to follow an evolutionary order, and then reverse the order. Furthermore, the title says humans, here mammals.
Section 8.2 ZntB: how does it transfer zinc? Outward or inward?
Reviewer 3 Report
Comments and Suggestions for Authors
The review presents results of intensive structural studies of Zn transporters. After short overview of the variety of Zn transporters in different groups of organisms, authors focus on the bacterial YiiP and human transporters ZnT7 and ZnT8. These systems work as Zn2+/proton antiporters, and belong to CDF family. Analysis of Cryo-EM and crystal structures of these transporters in different states of the reaction cycle, resulted in further development of the models of transport cycle proposed in structural studies. The switch between IF and OF states occur upon TM rearrangement and in the case of human homologs His-loop is involved in the recruiting of Zn ions from cytosol.
The review is clearly written and interesting to read. The proposed model is supported by structural studies and well illustrated.
Minor remarks
Line 104-105
Concerning homo- and heterodimeric transporters, the question is whether here are mentioned dimers composed of same or different subunits? Or the functional state of monomers can be similar or different within the dimer, just like it is discussed later in the text.
Line 98
...metal transporting proteins (MTPs) from plants.
Line 114
Metal tolerance proteins (MTPs)...
Line 176
...human ZnT7 (HsZnT7) (45), human ZnT8 (HsZnT8)... To what degree one can compare both human systems? from the section 3.1 follows that these systems belong to different groups.
Fig. 1C
One can add PDB codes on the panel, for example, above the structures
Fig. 1D
Violet shading is too dark, letters are unreadable. Also would be fine to group aligned sequences more or less into prokaryotic/bacterial, human, or probably plant, how it is discussed in the section 5.1.
Fig. 2
Why some sequences (left) are in bold? Also NgMTP1 is bold?
Fig. 3 A-C
One can add PDB ID (something like bact. YiiP IF: 3H90 and OF: 7KZZ) and make letters of corresponding color. Like it is done in E-G panels.
Fig. 3D
Which structure is shown here, probably HsZnT7?
Renumber Fig.5 to Fig.4?
Fig. 4
The legend to Panel D is absent.
Line 479
Undoubtedly, multiple ZnTs work co-operatively work to maintain Zn2+ homeostasis...
Reviewer 4 Report
Comments and Suggestions for Authors
The manuscript is very well-written, and the topic is also in the reader's demand and covers good pieces of information which is also helpful for the readers. Here are some minor revisions as following:
Line 2, Title: The paper is mainly focused on bacteria and humans' zinc transport but also discusses yeasts, fungi and plants' Zn transporters. Therefore, keep the title simple by writing "..... Zn transporters in different biological kingdoms" rather than "..... Zn transporters in bacteria, plants and humans.
Line 31: Provide examples of Zn-dependent enzymes and cite the following articles (this will provide added information confirming the essentiality of the Zn for intracellular metabolism):
Human liver alcohol dehydrogenase: https://link.springer.com/article/10.1007/s00018-008-8593-1
Human placenta alkaline phosphatase: https://onlinelibrary.wiley.com/doi/10.1002/ardp.202000011
Bacillus subtilis amylase: https://doi.org/10.1080/10826068.2018.1479863
Bacillus subtilis lipase: https://doi.org/10.1016/j.ijbiomac.2019.06.042
Bovine pancreas carboxypeptidase A: https://www.mdpi.com/2072-6651/12/11/680
And other Zn-dependent enzymes: https://doi.org/10.1016/B978-012095440-7/50039-1
Line 58-61: Provide the PDB ID of the mentioned Zn transports.
Line 74-76: Transport through uniport can also be active transport if it is against the concentration gradient. Correct your statement.
Line 82: Write the standard form of ZRT/IRT.
Line 87: ZntB of which organism?
Line 96: ZitB-like, ZnT1-like, Zrc1-like proteins of what organism?
Line 126: Zn transport of filamentous fungus should also be discussed in a few lines in section 3.3 (change the subheading to Yeast and fungal ZnTs) by citing the following articles:
https://doi.org/10.3389/fcimb.2013.00065
https://doi.org/10.3389/fmicb.2019.02251
https://link.springer.com/article/10.1007/s11274-023-03737-7
https://doi.org/10.3390/jof6040305
https://doi.org/10.1111/1462-2920.15542
Figure 1B: Zn2+ and H+ should be represented by 2 different colors. As said in line 84, ZnT exports Zn2+, so why in Fig 1B are both: Extracellular or intracellular compartments? Why H+ import is not shown in the first monomer? In CTD, alpha1 is 2 times, where is alpha 2 (check Figure 4B, where alpha2 is mentioned). Redraw Fig 1B and make clear every aspect of it.
Figure 1C: What is "A" in the figure representing (no description in legend)? How does this figure justify it is IF and OF?
Line 175, 177: There is no Table 1 in the manuscript.
Figure 2: Provide the name of the organisms/sequence IDs in the multiple sequence alignment.
Figure 2: How can different Zn transporters be categorised based on loop length in MSA? Label A, B and C on these three alignments.
Figure 5: To make it easy for readers: The simplified cartoon should come first as section A then, Zn coordination structures (B), and TM helix rearrangements (C).
Figure 4: (A) Why Zn 2+ ion is not shown with HsZnT7? (B) Provide organism names and sequence IDs in the alignment.
Figure 4D: the figure 4D is not described in the legend.
Line 231 and 269: Why does Fig 5 come first then Fig 4? Arrange the figure numbers in the sequence.
Figure 6: Why only bacterial YiiP mechanism for Zn2+ recruitment is discussed and why not human ZnT7 and 8? How YiiP is different/similar to ZnT7/8 in the mechanism?
ZIPs are not well discussed in the manuscript. How they are different from ZnTs.
Line 459: section 9 Conclusions should be bold.
Comments on the Quality of English LanguageOverall English is good.
